# Antioxidant Efficacy of a *Spirulina* Liquid Extract on Oxidative Stress Status and Metabolic Disturbances in Subjects with Metabolic Syndrome

**DOI:** 10.3390/md20070441

**Published:** 2022-07-01

**Authors:** N’Deye Lallah Nina Koite, N’gouro Issa Sanogo, Olivier Lépine, Jean-Marie Bard, Khadija Ouguerram

**Affiliations:** 1Département de Recherche en Santé Publique, Faculté de Pharmacie, Université des Sciences, des Techniques et des Technologies, Bamako J287+PM5, Mali; ndeyevictor@gmail.com; 2Institut Jérôme Lejeune, 75870 Paris, France; ngsanogo@gmail.com; 3AlgoSource Technologies, 44600 Saint Nazaire, France; olivier.lepine@algosource.com; 4Centre National de la Recherche Scientifique, ISOMer—UE 2160, IUML—Institut Universitaire Mer et Littoral, Nantes Université, 44035 Nantes, France; jean-marie.bard@univ-nantes.fr; 5Institut de Cancérologie de l’Ouest, 44805 Saint-Herblain, France; 6Centre de Recherche en Nutrition Humaine Ouest (CRNH-O), 44093 Nantes, France; 7Centre de Recherche en Nutrition Humaine Ouest (CRNH-O), Unité Mixte de Recherche, Institut des Maladies de l’Appareil Digestif (IMAD), NRAE, Physiopathologie des Adaptations Nutritionnelles (PhAN), Nantes Université, 44093 Nantes, France

**Keywords:** metabolic syndrome, dyslipidemia, oxLDL/LDL cholesterol, isoprostanes, *Arthrospira* liquid extract

## Abstract

Lipid peroxidation is associated with the development of some pathologies, such as cardiovascular diseases. Reduction in oxidative stress by antioxidants, such as *Arthrospira* (formely *Spirulina*), helps improving this redox imbalance. The aim of the study was to evaluate the effect of the *Arthrospira* liquid extract “Spirulysat^®^” on oxidative markers—in particular, oxidized LDL (oxLDL)/total LDL cholesterol—and isoprostanes and to investigate its impact on lipid and glucose metabolism in the metabolic syndrome subject. A controlled, randomised, double-blind design was conducted in 40 subjects aged 18 to 65 years with metabolic syndrome after a daily intake of Spirulysat^®^ or placebo for twelve weeks. Blood and urinary samples were collected at three visits (V1, V2, V3) in the two groups for parameters determination. Although the Spirulysat^®^ group showed a decrease at all visits of the oxLDL/total cholesterol ratio, there was no significant difference compared to the placebo (*p* = 0.36). The urinary isoprostanes concentration in the Spirulysat^®^ group was reduced (*p* = 0.014) at V3. Plasma triglycerides decreased at V3 (*p* = 0.003) and HDL-cholesterol increased (*p* = 0.031) at all visits with Spirulysat^®^. In conclusion, Spirulysat^®^ did not change the oxidized LDL (oxLDL)/LDL ratio but decreased the urinary isoprostanes, plasma triglycerides and increased HDL cholesterol, suggesting a beneficial effect on metabolic syndrome.

## 1. Introduction

Metabolic syndrome is related to cardio metabolic risk factors and lipid disorders [1]. Worldwide, cardiovascular diseases (CVD) are the leading causes of mortality and morbidity. It is expected that by 2030, mortality from CVD will reach 22.5 million people, compared with 17.5 million deaths in 2012 [2].

It has been reported that patients with metabolic syndrome have a decreased efficiency of their antioxidant defences and an increased level of protein and lipid oxidation. They also suffer from hyperglycaemia and elevated triglycerides, as well as reduced concentrations of HDL-C [3,4] and so, a higher risk of cardiovascular disease [5]. As lipids are a target of oxidative stress [6], the resulted modification of plasma low-density lipoproteins (LDL) to the oxidized form (ox-LDL) is strongly involved in the development of atherosclerosis [7]. Isoprostanes are also considered as reliable markers of oxidative stress [6,8], while high-density lipoprotein (HDL or HDL-c) are known to exert an antiatherogenic role, including an important antioxidative activity [9]. Many studies have reported the importance of the dietary antioxidant supplementation as a strategy to boost antioxidant defences [10,11].

A new area of research in pharmacy and bromatology on the beneficial effects of microorganisms have been reported by researchers due to various chemical, biological and nutritional properties, this is the case of *Arthrospira* (formely *Spirulina*) [12,13].

*Arthrospira platensis* of the phylum Cyanobacteria grows in aquatic ecosystems [14], and species of *Arthrospira* have been isolated from tropical waters to the North Sea [15]. It has been consumed by humans for centuries in different parts of the world [16]. Several *Arthrospira* varieties have been studied, the most studied being *Arthrospira platensis*, *Arthrospira maxima* and *Arthrospira fusiformis* [14]. *Arthrospira platensis*, which is the species studied in our study, is identified to have a low concentration of heavy metals [17] and could play a useful role in respecting the environment for bioremediation, nitrification and carbon dioxide (CO_2_) fixation [18].

*Arthrospira* through its various beneficial properties on health, including its antioxidant activity could be an effective remedy in the prevention of the metabolic syndrome [18,19,20]. Although the number of microalgae species in nature is estimated between 200,000 and 800,000, only a few are used in food applications [21]. Microalgae exploitation as a source of protein and other bioactive products in human nutrition still presents some drawbacks, mainly due to the poor development of targeted technologies and processes for microalgae processing. Spirulysat^®^ obtained from *Arthrospira* microalgae, is liquid extract concentrated in phycocyanin, a powerful antioxidant, and contained polysaccharides. In recent studies, we showed that a *Arthrospira* liquid extract (Spirulysat^®^) concentrated in phycocyanin, a powerful antioxidant, and polysaccharides, increases antioxidant defences in mice and hamsters submitted to a hypercaloric diet [22,23]. In high fat high sucrose fed mice, Spirulysat^®^ supplementation protects against non-alcoholic steatohepatitis [22]. We also reported that Spirulysat^®^ supplementation prevent metabolic syndrome-associated metabolic disturbances in hamsters [23] and atherosclerosis development in apolipoprotein E-deficient mice [24].

The aim of the present study was to evaluate the antioxidant efficacy of Spirulysat^®^ on plasma oxidized LDL levels and urinary isoprostanes concentration and then investigate lipid and glucose metabolism in human subjects with metabolic syndrome using a controlled, randomised, double-blind design.

## 2. Result

### 2.1. Demographics and Anthropometric Measurements of Patients with Metabolic Syndrome

The average age in the Spirulysat^®^ group was 51.8 ± 11.01 and in the placebo group 48.1 ± 8.52 with a male predominance of 55%. At inclusion at visit 0 (V0), no difference between groups was observed neither for waist circumference, nor BMI (Table 1), heart rate (bpm) or systolic (mmHg) and diastolic (mmHg) blood pressure (data not shown). Spirulysat^®^ supplementation did not induce any change in all these parameters.

### 2.2. Effects of Spirulysat^®^ on Oxidative Stress

Mean oxidized LDL cholesterol (oxLDL)/total LDL concentrations between Spirulysat^®^ and placebo group are presented in Table 2. No significant difference in the oxLDL/total LDL cholesterol ratio was found between the supplemented and placebo groups at any visits (*p* = 0.36 at V3). Spirulysat^®^ supplementation decreased urinary isoprostane at V3 compared to placebo (2.31 ± 0.91 vs. 3.51 ± 2.11, *p* = 0.014).

### 2.3. Effects of Spirulysat^®^ on Lipid and Glucose Metabolism

Mean triglycerides levels were significantly decreased in the Spirulysat^®^ group compared to placebo at V3 (1.23 ± 0.57 vs. 1.97 ± 0.80 g/L, *p* = 0.003). At V3, HDL cholesterol was significantly increased in the Spirulysat^®^ group compared to the placebo (0.55 ± 0.14 vs. 0.48 ± 0.18 g/L, *p* = 0.031). This increase was also observed at visit 2. Although glycemia showed an increase at all visits in Spirulysat^®^ group, the values were not significantly different. Insulinemia tended to be higher in Spirulysat^®^ group than in the placebo group at inclusion (*p* = 0.044); this difference was no longer found at the following visits (Table 2). Fatty liver index (FLI) did not show significant difference between the two groups. Nevertheless, the variation between visits 3 and visit 1 (V3–V1) (−3.48 ± 7.94 vs. 1.56 (13.47), *p* = 0.036) and between visit 3 and visit 2 (−3.35 (7.18) vs. −0.09 (12.38), *p* = 0.016) in Spirulysat^®^ is significantly decreased compared to placebo group. No significant change between visits was found for liver enzymes (data not shown).

**Table 2 marinedrugs-20-00441-t002:** Effect of Spirulysat^®^ supplementation on different clinical parameters in studied subjects.

Measured Variables	V1			V2			V3		
	Placebo n = 20	Spirulysat^®^ *n* = 20	*p*-Value	Placebo *n* = 20	Spirulysat^®^ *n* = 20	*p*-Value	Placebo *n* = 20	Spirulysat^®^ *n* = 20	*p*-Value
**oxLDL/LDL cholesterol ratio (U/g)**	67.44 ± 23.04	61.15 ± 20.53	0.58	72.22 ± 17.76	67.16 ±14.37	0.41	74.06 ± 16.08	69.08 ± 12.72	0.36
**Triglycerides (g/L)**	1.83 ± 0.82	1.35 ± 0.46	0.11	2.02 ± 0.89	1.58 ± 0.80	0.13	1.97 ± 0.80	1.23 ± 0.57	0.003
**LDL cholesterol (g/L)**	1.53 ± 0.36	1.52 ± 0.33	0.88	1.51 ±0.44	1.49 ± 0.32	0.96	1.43 ± 0.38	1.49 ± 0.33	0.22
**HDL cholesterol (g/L)**	0.47 ± 0.13	0.54 ± 0.09	0.004	0.47 ± 0.16	0.54 ± 0.11	0.012	0.48 ± 0.18	0.55 ± 0.14	0.031
**Total cholesterol (g/L)**	2.36 ± 0.46	2.33 ± 0.36	0.96	2.38 ± 0.50	2.34 ± 0.35	0.77	2.31 ± 0.43	2.30 ± 0.34	0.57
**Fatty Liver Index (** **FLI)**	72.95 ± 15.39	68.90 ± 19.89	0.71	74.25 ± 16.46	69.18 ± 20.36	0.48	75.32 ± 17.53	65.63 ± 22.16	0.19
**Glycemia (mmol/L)**	5.26 ± 0.64	5.56 ± 0.56	0.072	5.48 ± 0.76	5.55 ± 0.65	0.84	5.38 ± 0.76	5.90 ± 0.67	0.056
**Insulinemia (mU/L)**	10.06 ± 4.78	14.17 ± 8.11	0.044	11.39 ± 3.41	13.89 ± 6.60	0.33	11.95 ± 4.82	16.13 ± 10.88	0.28
**Urinary isoprostane (µg/24 h)**	2.86 ± 1.96	2.61 ±1.44	0.82	2.98 ± 1.56	2.20 ± 1.09	0.057	3.51 ± 2.11	2.31 ± 0.91	0.014
**ALAT (µkat/L)**	0.48 ± 0.31	0.59 ± 0.35	0.27	0.45 ± 0.23	0.57 ± 0.33	0.16	0.62 ± 0.58	0.57 ± 0.48	0.7
**ASAT (µkat/L)**	0.41 ± 0.14	0.43 ± 0.14	0.63	0.42 ± 0.13	0.42 ± 0.15	0.85	0.49 ± 0.30	0.43 ± 0.16	0.64
**GGT (U/L)**	50.22 ± 50.07	46.84 ± 34.97	>0.99	45.67 ± 37.75	45.23 ± 28.84	0.69	54.64 ± 59.04	49.07 ± 36.69	0.54

Missing data for fatty liver index (FLI) (*n* = 4; placebo = 1, Spirulysat^®^ = 3); glycemia (*n* = 1); insulinemia (*n* = 1); urinary isoprostane (Spirulysat^®^, *n* = 2); urinary isoprostane (Spirulysat, *n* = 2). Results are presented as Mean (SD); Wilcoxon rank sum test. oxLDL: oxidized low-density lipoprotein, LDL: low density lipoprotein, HDL: high density lipoprotein, ALAT: alanine aminotransferase, ASAT: aspartate aminotransferase, GGT (gamma-glutamyltransferases).

## 3. Discussion

This controlled, randomised, double-blind clinical trial has evaluated the antioxidant efficacy of 12 weeks of supplementation with a liquid extract of *Arthrospira* (Spirulysat^®^) rich in C-Phycocyanin (C-PC) and polysaccharides compared to a placebo on the level of oxidized LDL, glucose, lipid metabolism and 24-h urinary isoprostanes in subjects with metabolic syndrome. This trial did not show a significant effect of Spirulysat^®^ supplementation on the oxidized LDL (oxLDL)/LDL cholesterol ratio. Nevertheless, there was a significant decrease in the 24-h urinary isoprostanes concentration and the mean plasma triglycerides concentration and an increase in HDL cholesterol at all visits in the Spirulysat^®^ group compared to the placebo. We also measured a decrease in variation between visit 3 and visit 1 and between visit 3 and visit 2 of the fatty liver index in the Spirulysat^®^ group compared to placebo. 

Our study is the first to investigate the effect of liquid *Arthrospira* extract on oxidative stress, using a reliable marker, such as oxLDL and urinary isoprostanes, in subject with metabolic syndrome. Spirulysat is composed of several bioactive molecules, but the molecule with the highest concentration is C-PC followed, at a much lower level, by polysaccharides.

Phycocyanin composed of an apoprotein and a phycocyanobilin is light-sensitive and must be kept in darkness [25,26], and the participating subjects in the present trial were advised to use the ampoule-containing Spirulysat^®^ as soon as it is removed from its box. Oral administration of Spirulysat^®^ exposes phycocyanin to gastrointestinal proteolysis. Based on in vitro studies [27], C-PC is rapidly digested by pepsin in simulated gastric fluid. There are scarce literature data about the bioactivities of peptides obtained after phycocyanin digestion but we can suppose from this study [27] that the effects we observed in this trial are due to phycocyanobilin. Indeed it has also been reported that phycocyanin at a dose of 300 mg/kg administered orally for 10 weeks showed the same protective effect as oral administration of phycocyanobilin at a dose of 15 mg/kg for 2 weeks [28]. Spirulysat^®^ is a liquid extract of fresh *Arthrospira platensis* titrated in C-PC. This extract contains, in addition to phycocyanin C-PC, other water-soluble molecules, proteins, amino acids, enzymes, sugars, water-soluble vitamins and mineral salts. The observed results in the present trial are probably related to a synergic effect of these molecules and essentially to a predominant antioxidant component, the phycocyanin C-PC. It is well established that oxidative stress and inflammation are involved in the development of cardiovascular disease [7,29]. Phycocyanin, the main component of Spirulysat^®^, has antioxidant activity as it is able to scavenge various radicals and inhibit lipid peroxidation [30,31] and has anti-inflammatory properties [32,33]. We have shown that supplementation with Spirulysat^®^, which contains a high amount of C-PC, significantly decreases the 24-h urinary isoprostanes concentration, suggesting a better redox balance. Isoprostanes are considered to be specific markers of oxidative lipid damage in the body, and an overproduction of isoprostanes is linked to oxidative stress [8,34]. The decrease in isoprostanes in the present study is consistent with our previous study in mice, which reported a decrease in oxidative stress [22]. In the mice study, we observed a strong association between Spirulysat^®^ supplementation and antioxidant parameters, bile acid modification, impact on food intake and modulation of gene expression. Furthermore, numerous studies on human and animal models have reported the antioxidant effect of *Arthrospira* [22,35,36], and it was proved to be related essentially to C-PC [32,37]. It also appears that the antioxidant effect can also be linked to the presence of polysaccharides in the liquid extract isolated from *Arthrospira platensis* [38] and that these sulphated polysaccharides prevent many potential health risks, including cerebrovascular disease, cancer and chronic inflammation [39]. The effect of *Arthrospira* on urinary isoprostanes has not been reported in the literature in patients with metabolic syndrome but data from some studies [35,40] have shown a favourable effect of *Arthrospira* on oxidative stress and inflammation. [24]. A review from Spahis et al. enumerated several types of connection between metabolic disturbances and oxidative stress in metabolic syndrome [41]. Indeed, many studies showed that supplementation with antioxidant reverse metabolic disturbances, such as dyslipidaemia and insulin resistance [42,43]. In our study, the supplementation with Spirulysat^®^ induces a significant change on the lipid profile. The mean plasma triglycerides level decreased significantly in the Spirulysat^®^ group compared to the placebo, while HDL-cholesterol levels increased at all visits in this group. Thus, these results suggest a protective effect of Spirulysat^®^ against cardiovascular disease and are consistent with our data obtained in mice [24] and hamsters [23], fed hyperenergetic diets. Indeed, Spirulysat^®^ supplementation of apolipoprotein E-deficient mice, a model of a human atherosclerosis, during gestation and lactation, decrease atherosclerosis development in adult offspring [24]. Similarly, we showed in hamsters fed with a high-fat diet, that Spirulysat^®^ supplementation improves sphingolipids profile and protects from lipid accumulation in aorta, suggesting a protective effect against cardiovascular disease [23]. Using whole *Arthrospira,* several studies in humans [44,45] support the beneficial effects of *Arthrospira* in improving hyperlipidaemia and reducing risk factors for cardiovascular disease. Similarly, a meta-analysis, including twelve trials with a dose of *Arthrospira* ranging from 1 to 19 g/d over an intervention period of 2 to 48 weeks, reports a significant reduction in LDL-cholesterol, triglycerides and total cholesterol levels.

In this study, we were also interested in the effect of Spirulysat^®^ on glucose homeostasis. Spirulysat^®^ did not change glucose homeostasis, as assessed by fasting glucose and insulin concentration. In our previous studies in mice and hamsters, a higher dose of Spirulysat^®^ has improved glucose homeostasis [22,23]. It will be interesting to examine the effect of higher doses of Spirulysat^®^ in metabolic syndrome subjects exhibiting a lower glucose tolerance. Meta-analysis of eight trials reports a favourable effect of *Arthrospira* supplementation on fasting blood glucose and lipid profile [46]. Another meta-analysis of 12 trials reported a decrease in fasting glucose [47].

Finally, in the present study, we determined the effect of Spirulysat^®^ supplementation on fatty liver index. Although in the supplemented group we observed a significant difference between V3 and V1 and between V3 and V2 (data not shown), we measured no difference between the two groups (Spirulysat^®^ vs. placebo). This apparent contradiction is probably related to the small number of studied subjects and/or the small amount of Spirulysat^®^ consumed during the clinical trial. Indeed, using the same Spirulysat^®^ liquid extract but 10 times concentrated in C-PC, we have reported a significant protective effect on non-alcoholic steatohepatitis in mice submitted to a high-fat high-sucrose diet [22]. We also observed a lower lipid accumulation in the liver of hamsters fed a hyper-energetic diet supplemented with more concentrated Spirulysat^®^ [23]. It has been shown that the addition of whole *Arthrospira* to the diet plays a role in the decrease in liver fat and also a significant change in ALAT and ASAT [48]. In addition, animal studies have reported that the hepatoprotective activities of *Arthrospira* are associated with its antioxidant and anti-inflammatory components (C-phycocyanin, β-carotene and vitamin E) and with the reduction in the liver lipid profile [40,49].

Finally, Spirulysat^®^ supplementation for 12 weeks did not induce any side effects or increase in biological parameters measured, such as ALAT and ASAT, compared to the placebo group. None of the subjects stopped the study prematurely due to an adverse event. Hemodynamic measurements appeared stable throughout the study in both groups and in the entire study population. Using a higher dose of Spirulysat^®^ in hamsters for 12 weeks and in mice for 25 weeks, we did not observe deleterious effects on biological parameters or mortality [22,24]. No adverse effects are reported in previous studies using higher doses of phycocyanin, notably in mice [32,50] and in rats [51]. Thus, the safety profile of the products during the study can be considered as good. However, it will be necessary to confirm the safety of Spirulysat^®^ in humans over a longer period with studied dose or higher.

## 4. Materials and Methods

### 4.1. Preparation of Arthrospira Liquid Extract (Spirulysat^®^)

The *Arthrospira* extract used in the SPIROX study is a water extract obtained without any chemical solvents and using only mechanical devices. The extraction and formulation process are made at cold temperatures (15 °C) to preserve the active molecules of the extract, in particular phycocyanin. The main components are protein 2 g/L of which phycocyanin is 1 g/L (50%) and polysaccharide constitute 0.5 g/L. Spirulysat^®^ contains, among other components, phycocyanin, polysaccharides and other molecules, such as proteins, amino acids, enzymes, water-soluble vitamins and mineral salts.

The product is standardized using spectrophotometry thanks to the blue colour of phycocyanin. The production process can be described as follows. First, *Arthrospira* is cultivated in controlled conditions in a greenhouse, using the Algosource based culture medium. Both pH and temperature are daily monitored. The *Arthrospira* strain cultivated is the PCC 8005 from the Pasteur Institute in France. Fresh water is taken from the French national network and is allowed for human consumption. All nutrients involved are quality controlled. This stage is the purpose of a patent “Production process for micro algae” (Patent FR 92-11877).

The quality control is performed using microscopic analysis and Spectrophotometry on a daily basis. *Arthrospira* is harvested by filtration; each batch is controlled, and the following parameters are recorded: production unit, harvesting date, dry weight. The *Arthrospira* biomass is then frozen. In order to perform the water-based extraction, the *Arthrospira* biomass has been thawed, and after de-freezing *Arthrospira* cells are broken using centrifugation and water as the solvent. Hydrophilic compounds are then separated from lipids and cell residues using membrane filtration. This stage is covered by the patent N°EP3601319. Finally, the liquid extract is sterilized by micro filtration at 0.2 microns (Sartorius filter) and controlled with Spectrophotometry. It goes through a sterilizing filter into the final packaging: ampoules or sterile bag. Analysis for heavy metal detection and measurement and various nutrients (Table 3) are performed by a certified laboratory, Eurofins (www.eurofins.fr, accessed on 27 June 2018). The batch used for this clinical trial was subject to the same quality requirements and procedures as the industrial batches. Active products were prepared by the company Alpha Biotech—La Frostidié—44410 Asserac—France, according to good manufacturing practices. This food supplement Spirulysat^®^ has been marketed in France since 2012.

### 4.2. Study Design

#### 4.2.1. Pre-study Recommendations

Participants were given the following instructions: not to change lifestyle habits during the study (physical activity, smoking and alcohol consumption), not to do strenuous exercise during the 2 days preceding each visit, their eating habits during the study (no special diet, no taking of food supplements, no heavy meals or alcohol abuse), not to donate blood chronic drug treatments for those authorized (active and dosage) and not to take any new drugs during the study (having an impact on the study parameters), except in extreme cases.

#### 4.2.2. Study Design

The study was designed as a randomized, double-blind, placebo-controlled, two-groups parallel. Participants were randomly assigned to one of two groups: Spirulysat^®^ or placebo (Appendix A). Spirulysat^®^, as well as placebo, are packaged in 10 mL vials. The comparative product (placebo) exhibits the same characteristics, appearance, packaging and composition as the Spirulysat^®^, except that the product is replaced by a classic blue food colouring (Rainbow Dust, Colour Flo, Preston, UK).

C-Phycocyanin is the main pigment of *Arthrospira*. The Food and Drug Administration (FDA) has granted *Arthrospira* the status of GRAS (Generally Recognized as Safe (GRAS Notice No. GRN 000391). Indeed, phycocyanin is used as blue dye in the formulation of food products, such as desserts and sweets. According to the EFSA’s Scientific Opinion, 2010, the daily dose recommended to produce a claim effect is 2 g daily of *Arthrospira*. This corresponds to an amount of 200–300 mg of Phycocyanin. Thus, in the present study we chose to investigate a low dose of about 20 mg per day per person (2 × 10 mg). Each randomized subject received 10 boxes of 20 vials with the prescription of 2 vials to be consumed per day (morning, just before breakfast) of Spirulysat^®^ or placebo for 12 weeks. As this study is the first clinical trial whose aim is to investigate the effect of Spirulysat^®^ on the oxidative status, we based the design of the present trial on the hamster data that showed a preventive effect of Spirulysat on metabolic syndrome. Indeed, hamster supplementation for 12 weeks prevents the perturbation of some metabolic parameters characteristic of the metabolic syndrome. Thus, we hypothesised that 12 weeks might be sufficient to see an effect on at least one component of the metabolic syndrome. Moreover, the choice of taking the Spirulysat at breakfast was done without any supporting scientific data, and only so that all participants take it at the same time to avoid any variability related to the time of intake. Subjects included in the study were scheduled for a screening visit (V0) followed by an inclusion visit (V1) that took place 1 to 2 weeks after V0. An intermediate follow-up visit (V2) was scheduled after 6 weeks of intervention and the end-of-study visit was conducted after 12 weeks of supplementation (V3) (Figure 1). This randomized clinical protocol was approved by the Ethics Committee protection (CPP) of Ouest of Rennes. It is registered under the number PEC15039 and NTC 02817620, respectively, in CCP and clinicaltrials.gov. A written informed consent was obtained from each subject.

### 4.3. Study Participants

The clinical trial was conducted in 2016 by Biofortis Mérieux NutriSciences. The flowchart of the study is shown in Figure 2. A total of 91 participants with metabolic syndrome were registered and finally 40 participants (22 men, 18 women, 20 per group) were enrolled in the present study. Inclusion criteria were: age between 18 and 65 years, BMI between 25 and 35 kg/m^2^, with a metabolic syndrome defined as central obesity, waist circumference >94 cm for men and >80 cm for women associated with at least 2 observed criteria (fasting blood triglycerides >1.5 g/L, fasting HDL cholesterol <0.4 g/L for men and <0.5 g/L for women, fasting blood glucose >1 g/L, blood pressure >130/85 mmHg) or under antihypertensive treatment, non-smoker or with tobacco consumption <10 cigarettes/day, for non-menopausal women with the same reliable contraception for at least three months prior to the start of the study and committed to maintaining it for the duration of the study or menopausal women without or with hormone replacement therapy started at stable dose. Ninety-one subjects were identified for the study and fifty-one were eventually not selected since they did not meet one or more of the eligible criteria inclusion for the study (assessment based on medical examination at V0 with verification at V1). In the clinical trial, subjects belonging to these categories are not included: suffering from a metabolic disorder, such as diabetes or uncontrolled thyroid disorder (E1), suffering from a serious chronic disease (cancer, HIV, renal failure, ongoing liver or biliary disorders, chronic inflammatory digestive disorders) or gastrointestinal disorders (coeliac disease) (E2), with a history of ischemic cardiovascular event (E3), having undergone recent surgical procedure, less than 6 month (E4), suffering from uncontrolled hypertension (systolic blood pressure ≥ 160 mmHg and/or diastolic blood pressure ≥ 100 mmHg) (E5), regular intake of dietary supplements or “functional foods” impacting on lipid metabolism or stopped less than 3 months prior to the V0 visit (E6), with a significant change in dietary habits or physical activity in the 3 months prior to the V0 visit or not agreeing to comply (E7), fasting blood triglycerides > 3.5 g/L (3.95 mmol/L) (E8) and blood hsCRP > 10 mg/L (E9).

After the screening period, each subject was randomly assigned to either Spirulysat^®^ or a placebo. The random product attribution was performed after checking the subjects’ eligibility once the results were available for inclusion (after V0 screening visit), thus minimizing the selection bias. Products’ allocation depended only on the subjects’ inclusion sequence in the study.

### 4.4. Anthropometric and Hemodynamic Measurements

Body weight and waist circumference were measured at each visit from selection to the end of the study (4 visits: V0, V1, V2, and V3). Heart rate (bpm) and systolic (mmHg) and diastolic (mmHg) blood pressure were performed twice, separated by at least 2 min. The first measurement was performed after a minimum rest of 5 min.

### 4.5. Biological Parameters Measurements

All biological parameters were analysed in a central laboratory (Biofortis Mérieux Nutrisciences). Blood samples were taken after a 12-h fasting period. Urine was collected for 24-h and after mixing a sample was taken for analysis. The first one at visit 0 (screening), the second one at visit 1 (inclusion from 7 to 15 days after V0), the third one V2 (follow-up visit 15 ± 3 after V1), the fourth one V3 end of study visit (15 ± 3 days after V2). The blood is collected in different tubes depending on the dosage to be performed. For glucose determination the Fluor tubes were used and analysed by enzymatic Hexokinase/Roche diagnostic. Dry serum tubes were used for lipids determination. Concentration of total cholesterol and triglycerides were determined using enzymatic kit (Cobas integra 400+) and for HLD-c by photometry (Roche Diagnostic, Basel, Switzerland). The liver markers alanine aminotransferase (ALAT), aspartate aminotransferase (ASAT) and gamma-glutamyltransferases (GGT) were analysed by UV test (Cobas integra 400+/Roche Diagnostic, Basel, Switzerland), and GGT by enzymatic (Roche Diagnostic, Basel, Switzerland). LDLc was calculated according to Friedewald formula, and LDLox by ELISA technique. For the measure of urinary isoprostane, samples were collected 24-h prior to each visit (V1, V2, V3) and the assay was performed only on the 24-h urine to take into account the difference in the dilution 328 of participant’s urine. The urine was homogenised before aliquoting, then each aliquot was treated with Butylhydroxytoluene (BHT) (1 mL of urine with 10 µL of 0.025 M BHT solution) before freezing for analysis. Urinary isoprostane (isoprostane F2 alpha) was analysed by ELISA (Oxford Biomedical Research). The raw results from the laboratory were expressed in ng/mL, and then converted to the concentration of the whole 24 h collected urine (µg/24 h).

The oxLDL/total LDLc ratio (U/g) was calculated using the following formula:

OxLDL/LDL ratio (U/g) = oxLDL (U/L) /LDL (g/L). To assess the effect of Spirulysat^®^ on the liver lipids content, we calculated the fatty liver index (FLI) a validated parameter in diagnosing the presence of liver steatosis and its severity [52]. This index was calculated using the following formula:

FLI = (e0.953 × loge (triglycerides) + 0.139 × BMI + 0.718 × loge, GGT (gamma-glutamyltransferase) + 0.053 × waist circumference − 15.745)/(1 + e0.953 × loge (triglycerides) + 0.139 × BMI + 0.718 × loge (GGT) + 0.053 × waist circumference − 15.745) × 100, where triglycerides, GGT, wait circumference are expressed in mg/dL, U/L and Cm, respectively.

### 4.6. Statistical Analysis

The sample size was calculated according to the guidelines [53,54,55,56,57]. Statistical analysis was performed with R software version 4.1.2. Quantitative data are presented as mean ± SD (standard deviation of the mean). The comparison of continuous variables between groups at different visits was performed by independent sample Wilcoxon rank tests. To estimate the variation in fatty liver index (FLI) between visits (V3–V1 and V3–V2), Wilcoxon independent sample rank tests were performed. A *p* value of less than 0.05 was considered statistically significant.

## 5. Conclusions

In conclusion, Spirulysat^®^ did not change the oxidized LDL (oxLDL)/LDL ratio but decreased urinary isoprostanes and plasma triglycerides and increased HDL cholesterol, suggesting a beneficial effect on metabolic syndrome. We did not measure significative differences in oxLDL, which is also a good marker of oxidative stress. This could be related to the small number of subjects, which is low given the high inter-individual variability. To reinforce these data, and in particular those showing a trend with Spirulysat^®^ supplementation, it would be interesting, in a future study, to measure the oxLDL, urinary isoprostanes and 8-Hydroxyguanosine/8-oxo-2’-désoxyguanosine ratio in a larger number of subjects using Spirulysat^®^ more concentrated in C-PC.

## Figures and Tables

**Figure 1 marinedrugs-20-00441-f001:**
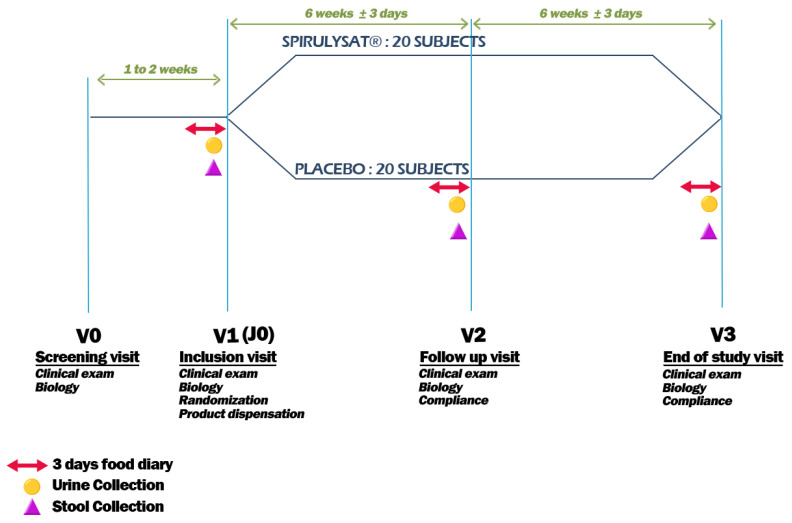
Study design.

**Figure 2 marinedrugs-20-00441-f002:**
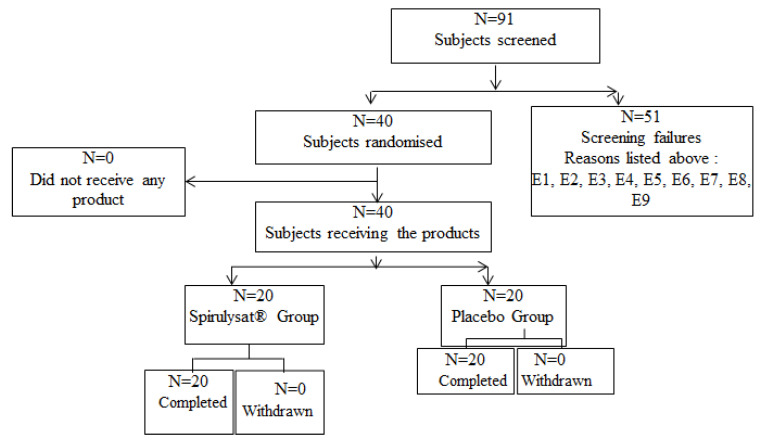
Flowchart study.

**Table 1 marinedrugs-20-00441-t001:** Characteristics of study population at the inclusion visit.

Variables	Placebo (*n* = 20)	Spirulysat^®^ (*n* = 20)
Age (years)	48.1 ± 11.01	51.8 ± 8.52
Gender:		
Male (n, %)	11 (55.0)	11 (55.0)
Female (n, %)	9 (45.0)	9 (45.0)
BMI (kg/m^2^)	29.73 ± 2.74	29.65 ± 2.72
Waist circumference (cm)	100 ± 7	100 ± 6
Triglycerides (g/L)	1.83 ± 0.82	1.35 ± 0.46
Total cholesterol (g/L)	2.36 ± 0.46	2.33 ± 0.36
LDL cholesterol (g/L)	1.53 ± 0.36	1.52 ± 0.33
HDL cholesterol (g/L)	0.47 ± 0.13	0.54 ± 0.09

**Table 3 marinedrugs-20-00441-t003:** Composition of Spirulysat^®^ for 100 mL.

	Amount in 100 mL
Proteins	˃0.2 g
Carbohydrate	0.05 g
Lipids	Traces
Vitamin B12	0.3 µg
Iron	0.2 mg
Magnesium	30 mg
Calcium	40 mg
Potassium	8 mg
Sodium	20 mg
Copper	30 µg
Zinc	20 µg
Phycocyanine	110 mg

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
