# Peer review of "Antioxidant Efficacy of a Spirulina Liquid Extract on Oxidative Stress Status and Metabolic Disturbances in Subjects with Metabolic Syndrome"

_marinedrugs, 2022, doi:10.3390/md20070441_

Round 1
Reviewer 1 Report
The authors have studied the anti-oxidative effect of aqueous extract of spirulina on the patients with metabolic syndrome. This is a clinical trial study and based on the previous work on the mice. The manuscript is well written. I have following comments to the authors
1. The authors didnot clearly define the metabolic syndrome patients in the method section. The exclusion criteria is well mentioned but what is the actual composition of group. The authors should provide a table for it.
2. The authors didnot mention the dose of Spirulysat®. How did the authors determine the dose of Spirulysat®?
3. What is the metabolic stability of the Spirulysat® in the patients?
4. The study period is 21 days which is a small time period and the authors have mostly used the blood parameters to come to a conclusion. The authors should provide a justification on it.
4. The long term demerits of Spirulysat®wasnot discussed in the discussion section.
5. The authors didnot discuss the mechanism of action of Spirulysat in this study.
Author Response
Together with the co-authors, we thank the reviewers for their careful reading and constructive comments. We present here the revised form of our manuscript entitled “Antioxidant efficacy of a spirulina liquid extract on oxidative stress status and metabolic disturbances in subjects with metabolic syndrome”.
We have taken all the reviewers' comments into account and believe that the revised version is significantly improved and will be of interest to readers.
Reviewer 1
1.The authors didnot clearly define the metabolic syndrome patients in the method section. The exclusion criteria is well mentioned but what is the actual composition of group. The authors should provide a table for it.
After the screening period each subject was randomly assigned to either Spirulysat® or a placebo. The random product attribution was performed after checking the subject's eligibility once the results were available for inclusion (after V0 screening visit), thus minimizing the selection bias. Product's allocation depended only on the subjects’ inclusion sequence in the study.
Table 4. Composition of the group (please see the attachment)
This explicative paragraph as well as the table showing the composition of group were introduced in the revised manuscript Line 307 to Line 311.
We also added the diagram representing the different phases of the study design (response to question 4)
- The authors didnot mention the dose of Spirulysat®. How did the authors determine the dose of Spirulysat® ?
C-Phycocyanin is the main pigment of spirulina. The Food and Drug Administration (FDA) has granted spirulina the status of GRAS (Generally Recognized as Safe (GRAS Notice No. GRN 000391). Indeed, Phycocyanin is used as blue dye in the formulation of food products such as desserts and sweets. According to the EFSA's Scientific Opinion, 2010, the daily dose recommended to produce a claim effect is 2 grams daily of spirulina which correspond to an amount of 200-300 mg of Phycocyanin. Thus in the present study we have chosen to test a low dose of 20 mg per person per day (2*10 mg).
Line 247 to line 252
- What is the metabolic stability of the Spirulysat® in the patients ?
C-Phycocyanine composed of an apoprotein and a phycocyanobilin is light sensitive and must be kept in darkness [1,2] and the participating subjects in the present trial were advised to use the vials containing Spirulysat as soon as they were removed from their box. Oral administration of Spirulysat exposes phycocyanin to gastro intestinal proteolysis. Based on in vitro study [3] ,phycocyanin is rapidly digested by pepsin in simulated gastric fluid. There are scarce literature data about bioactivities of peptides obtained after phycocyanin digestion but we can suppose from this same study [3] that the effect we observed in this trial is due to phycocyanobilin. Indeed, it has been reported that phycocyanin at the dose of 300 mg/kg given orally for 10 weeks showed the same protective effect than the oral administration of phycocyanobilin at the dose of 15 mg/kg for 2 weeks [4] . Line 139 to line 146
- The study period is 21 days which is a small time period and the authors have mostly used the blood parameters to come to a conclusion. The authors should provide a justification on it.
As this study is the first clinical trial whose aim is to investigate the effect of Spirulysat® on the oxidative status, we based the design of the present trial on Hamster data that showed a preventive effect of Spirulysat on metabolic syndrome. Indeed, Hamster supplementation during 12 weeks prevents the perturbation of some metabolic parameters characterising the metabolic syndrome. Thus we hypothesized that 12 weeks might be sufficient to see an effect on at least one component of the metabolic syndrome in subjects with this syndrome. So this trial was carried out over a period of 12 weeks with an inclusion period of 10 to 11 weeks or ± 44 days. There was a screening visit (V0) followed by an inclusion visit (V1) which took place 1 to 2 weeks after V0. The experimental phase with placebo or Spirulysat® then lasted 12 weeks. An intermediate follow-up visit (V2) was scheduled after 6 weeks of intervention and the end-of-study visit was conducted after 12 weeks of supplementation (V3).
We have added an explicative paragraph and a new figure (Figure 1) showing the different stages of the design. Line 253 to line 259
Figure 1 : Study design Line 268 to Line 285
- The long-term demerits of Spirulysat®was not discussed in the discussion section.
Spirulysat® supplementation for 12 weeks did not induce any side effects or increase in biological parameters measured such as ALAT and ASAT compared to the placebo group. None of the subjects stopped the study prematurely due to an adverse event. Hemodynamic measurements appeared stable throughout the study in both groups and in the entire study population. Using a higher dose of Spirulysat® in hamsters for 12 weeks and in mice for 25 weeks, we did not observe deleterious effects on biological parameters or mortality [5,6]. It is also reported in previous studies using higher doses of phycocyanin, notably in mice [7,8] and in rats [9]. Thus the safety profile of the products during the study can be considered as good. However, it will be necessary to confirm the safety of Spirulysat® in humans over a longer period and at the used or higher dose.
Line 197 to line 204
- The authors didnot discuss the mechanism of action of Spirulysat® in this study.
Spirulysat® is a liquid extract of fresh Spirulina (Arthrospira platensis) titrated in phycocyanin. This extract contains, in addition to phycocyanin, other water-soluble molecules, proteins, amino acids, enzymes, sugars, water-soluble vitamins and mineral salts. The observed results in the present trial are probably related to a synergic effect of these molecules and essentially to a predominant antioxidant component, the phycocyanin. It is well established that oxidative stress and inflammation are involved in the development of cardiovascular disease [10,11]. Phycocyanin, the main component of Spirulysat® exhibits an antioxidative activity because it is able to scavenge various radicals and to inhibit lipid peroxidation [12,13]. In addition, phycocyanin has anti-inflammatory properties [7,14]. In our previous studies in Hamster and mice we observed a strong association between Spirulysat® supplementation and antioxidant parameters, bile acid modification, impact on food intake, modulation of gene expression. Since in the present trial we measured a decrease in urinary isoprostanes, we can suggest that this antioxidative effect observed with Spirulysat® leads to ameliorating lipids profile (decrease in plasma triglycerides and increase in HDL cholesterol) we have measured.
These precisions are introduced in the discussion section, Line 146 to Line 158 and Line 161 to Line 163
References
- Benedetti, S.; Rinalducci, S.; Benvenuti, F.; Francogli, S.; Pagliarani, S.; Giorgi, L.; Micheloni, M.; D’Amici, G.M.; Zolla, L.; Canestrari, F. Purification and Characterization of Phycocyanin from the Blue-Green Alga Aphanizomenon Flos-Aquae. J. Chromatogr. B Analyt. Technol. Biomed. Life. Sci. 2006, 833, 12–18, doi:10.1016/j.jchromb.2005.10.010.
- Wang, L.; Qu, Y.; Fu, X.; Zhao, M.; Wang, S.; Sun, L. Isolation, Purification and Properties of an R-Phycocyanin from the Phycobilisomes of a Marine Red Macroalga Polysiphonia Urceolata. PLoS ONE 2014, 9, e87833, doi:10.1371/journal.pone.0087833.
- Minic, S.L.; Stanic-Vucinic, D.; Mihailovic, J.; Krstic, M.; Nikolic, M.R.; Cirkovic Velickovic, T. Digestion by Pepsin Releases Biologically Active Chromopeptides from C-Phycocyanin, a Blue-Colored Biliprotein of Microalga Spirulina. J. Proteomics 2016, 147, 132–139, doi:10.1016/j.jprot.2016.03.043.
- Zheng, J.; Inoguchi, T.; Sasaki, S.; Maeda, Y.; McCarty, M.F.; Fujii, M.; Ikeda, N.; Kobayashi, K.; Sonoda, N.; Takayanagi, R. Phycocyanin and Phycocyanobilin from Spirulina Platensis Protect against Diabetic Nephropathy by Inhibiting Oxidative Stress. Am. J. Physiol. Regul. Integr. Comp. Physiol. 2013, 304, R110-120, doi:10.1152/ajpregu.00648.2011.
- Coué, M.; Tesse, A.; Falewée, J.; Aguesse, A.; Croyal, M.; Fizanne, L.; Chaigneau, J.; Boursier, J.; Ouguerram, K. Spirulina Liquid Extract Protects against Fibrosis Related to Non-Alcoholic Steatohepatitis and Increases Ursodeoxycholic Acid. Nutrients 2019, 11, 194, doi:10.3390/nu11010194.
- Coué, M.; Croyal, M.; Habib, M.; Castellano, B.; Aguesse, A.; Grit, I.; Gourdel, M.; Billard, H.; Lépine, O.; Michel, C.; et al. Perinatal Administration of C-Phycocyanin Protects Against Atherosclerosis in ApoE-Deficient Mice by Modulating Cholesterol and Trimethylamine-N-Oxide Metabolisms. Arterioscler. Thromb. Vasc. Biol. 2021, 41, e512–e523, doi:10.1161/ATVBAHA.121.316848.
- Romay, C.; Armesto, J.; Remirez, D.; González, R.; Ledon, N.; García, I. Antioxidant and Anti-Inflammatory Properties of C-Phycocyanin from Blue-Green Algae. Inflamm. Res. Off. J. Eur. Histamine Res. Soc. Al 1998, 47, 36–41, doi:10.1007/s000110050256.
- Ou, Y.; Lin, L.; Pan, Q.; Yang, X.; Cheng, X. Preventive Effect of Phycocyanin from Spirulina Platensis on Alloxan-Injured Mice. Environ. Toxicol. Pharmacol. 2012, 34, 721–726, doi:10.1016/j.etap.2012.09.016.
- Gupta, M.; Dwivedi, U.N.; Khandelwal, S. C-Phycocyanin: An Effective Protective Agent against Thymic Atrophy by Tributyltin. Toxicol. Lett. 2011, 204, 2–11, doi:10.1016/j.toxlet.2011.03.029.
- Jürgens, G.; Hoff, H.F.; Chisolm, G.M.; Esterbauer, H. Modification of Human Serum Low Density Lipoprotein by Oxidation--Characterization and Pathophysiological Implications. Chem. Phys. Lipids 1987, 45, 315–336, doi:10.1016/0009-3084(87)90070-3.
- Libby, P. Inflammation in Atherosclerosis. Arterioscler. Thromb. Vasc. Biol. 2012, 32, 2045–2051, doi:10.1161/ATVBAHA.108.179705.
- Romay, C.; Ledón, N.; González, R. Further Studies on Anti-Inflammatory Activity of Phycocyanin in Some Animal Models of Inflammation. Inflamm. Res. Off. J. Eur. Histamine Res. Soc. Al 1998, 47, 334–338, doi:10.1007/s000110050338.
- Thangam, R.; Suresh, V.; Asenath Princy, W.; Rajkumar, M.; Senthilkumar, N.; Gunasekaran, P.; Rengasamy, R.; Anbazhagan, C.; Kaveri, K.; Kannan, S. C-Phycocyanin from Oscillatoria Tenuis Exhibited an Antioxidant and in Vitro Antiproliferative Activity through Induction of Apoptosis and G0/G1 Cell Cycle Arrest. Food Chem. 2013, 140, 262–272, doi:10.1016/j.foodchem.2013.02.060.
- Romay, C.; González, R.; Ledón, N.; Remirez, D.; Rimbau, V. C-Phycocyanin: A Biliprotein with Antioxidant, Anti-Inflammatory and Neuroprotective Effects. Curr. Protein Pept. Sci. 2003, 4, 207–216, doi:10.2174/1389203033487216.

Reviewer 2 Report
In this study, the authors evaluated the antioxidant efficacy of Spirulysat® on plasma oxidized LDL levels and urinary isoprostanes concentration and then investigated glucose and lipid metabolism in human subjects with metabolic syndrome using a controlled, randomised, double-blind design. The authors concluded that subjects with metabolic syndrome supplemented with Spirulysat® have a lower urinary isoprostanes concentration, suggesting a decrease in oxidative stress, and an improvement in the lipid profile was observed in this study as well as a decrease of fatty liver index with Spirulysat® supplementation.
Comments
The reviewer has some concerns as follows:
1. One of the major concerns is that the sample size is too small, which may affect the validity of the conclusions of this study due to limited data.
2. How to detect both urinary isoprostane (ng/mL) and urinary isoprostane (μg/24h)? How to collect the urinary samples in these two conditions? Why only data for urinary isoprostane (μg/24h) in Spirulysat group have significant difference compared to Placebo in V3? It should be clearly described in the Methods and discussed in the Discussion.
3. Some descriptions in the Conclusions section are improper. The descriptions for conclusion in the Abstract (Spirulysat® did not change the oxidized LDL (oxLDL)/LDL ratio but decreased urinary isoprostanes, plasma triglycerides and increased HDL cholesterol, suggesting a beneficial effect on metabolic syndrome) are more proper. Moreover, the limitations of this study can be clearly described in the Conclusions section.
Author Response
Together with the co-authors, we thank the reviewers for their careful reading and constructive comments. We present here the revised form of our manuscript entitled “Antioxidant efficacy of a spirulina liquid extract on oxidative stress status and metabolic disturbances in subjects with metabolic syndrome”.
We have taken all the reviewers' comments into account and believe that the revised version is significantly improved and will be of interest to readers.
Reviewer2
1.One of the major concerns is that the sample size is too small, which may affect the validity of the conclusions of this study due to limited data.
This criticism is very relevant and we will take it into account in a second planned trial. However, in the present study, the sample size is determined according to the various guidelines for choosing an appropriate sample size for a pilot study that have been published, ranging from 10 to 40 subjects per group [1–6]. It was therefore decided to include 20 subjects per treatment group in the present study, considering a low early drop-out rate usually observed in this type of study.
In the conclusion of the manuscript we have highlighted this weakness and the fact that another test with more subjects is needed. Line 353 to line 356
- How to detect both urinary isoprostane (ng/mL) and urinary isoprostane (μg/24h)? How to collect the urinary samples in these two conditions? Why only data for urinary isoprostane (μg/24h) in Spirulysat® group have significant difference compared to Placebo in V3? It should be clearly described in the Methods and discussed in the Discussion.
All urine samples were collected 24 hours prior to each visit (24 hours prior to visit V1, 24 hours prior to visit V2 and 24 hours prior to visit V3) and the assay was performed only on the 24-hour urine. On receipt the urine was homogenized before aliquoting, then each aliquot was treated with Butylhydroxytoluene (BHT) (1mL of urine with 10µL of 0.025M BHT solution) before freezing and analysis. The raw results from the laboratory are obtained in ng/mL and then converted to the concentration in the whole 24h collected urine.
The absence of a significant difference when the values are expressed in ng/ml is certainly linked to a difference in the dilution of the participants' urine, which is corrected by considering the total volume of urine of the 24 hours. As the value in ng/mL of urinary Isoprostanes does not provide additional specific information, we propose to remove it and keep only the concentration expressed per 24h.
To respond to reviewer comment we have introduced in the “methodes” section the following paragraph « All urine samples were collected 24 hours prior to each visit (24 hours prior to visit V1, 24 hours prior to visit V2 and 24 hours prior to visit V3) and the assay was performed only on the 24-hour urine. The collected urine was homogenised before aliquoting, then each aliquot was treated with Butylhydroxytoluene (BHT) (1mL of urine with 10µL of 0.025M BHT solution) before freezing and analysis. The raw results from the laboratory are expressed in ng/mL and then converted to the concentration in the whole 24h collected urine », Line 326 to line 332.
We also introduced in discussion section this sentence « 24 hour urinary isoprostanes line 130
to take into account the diffrence in the dilution of participant’s urine »,
- Some descriptions in the Conclusions section are improper. The descriptions for conclusion in the Abstract (Spirulysat® did not change the oxidized LDL (oxLDL)/LDL ratio but decreased urinary isoprostanes, plasma triglycerides and increased HDL cholesterol, suggesting a beneficial effect on metabolic syndrome) are more proper. Moreover, the limitations of this study can be clearly described in the Conclusions section.
In conclusion Spirulysat® did not change the oxidized LDL (oxLDL)/LDL ratio but decreased urinary isoprostanes, plasma triglycerides and increased HDL cholesterol, suggesting a beneficial effect on metabolic syndrome. We did not measure significative differences in oxLDL, which is also a good marker of oxidative stress. This could be related to the small number of subjects, which is low given the high inter-individual variability. To reinforce these data and in particular those showing a trend with Spirulysat® supplementation, it would be interesting, in a future study, to measure oxLDL, urinary isoprostanes and 8-Hydroxyguanosine/ 8-oxo-2'-désoxyguanosine ratio in a larger number of subjects using Spirulysat® more concentrated in C-PC.
These precisions appear line 348 to line 354
References :
- Birkett, M.A.; Day, S.J. Internal Pilot Studies for Estimating Sample Size. Stat. Med. 1994, 13, 2455–2463, doi:10.1002/sim.4780132309.
- Browne, R.H. On the Use of a Pilot Sample for Sample Size Determination. Stat. Med. 1995, 14, 1933–1940, doi:10.1002/sim.4780141709.
- Julious, S.A. Sample Size of 12 per Group Rule of Thumb for a Pilot Study. Pharm. Stat. 2005, 4, 287–291, doi:10.1002/pst.185.
- Hertzog, M.A. Considerations in Determining Sample Size for Pilot Studies. Res. Nurs. Health 2008, 31, 180–191, doi:10.1002/nur.20247.
- Sim, J.; Lewis, M. The Size of a Pilot Study for a Clinical Trial Should Be Calculated in Relation to Considerations of Precision and Efficiency. J. Clin. Epidemiol. 2012, 65, 301–308, doi:10.1016/j.jclinepi.2011.07.011.
- Teare, M.D.; Dimairo, M.; Shephard, N.; Hayman, A.; Whitehead, A.; Walters, S.J. Sample Size Requirements to Estimate Key Design Parameters from External Pilot Randomised Controlled Trials: A Simulation Study. Trials 2014, 15, 264, doi:10.1186/1745-6215-15-264.

Reviewer 3 Report
The manuscript submitted by Ouguerram Khadija and co-workers presents a set of analysis discussing the effects of the spirulina liquid on individuals suffering from metabolic syndrome.
The paper is very interesting. Introduction is brief and coherent, and gives a very good input into the biochemical background of metabolic syndrome, taking into account the current state of the art regarding dietary antioxidant supplementation.
The methods applied are described with sufficient amount of detail. The study design is very clear, supported by the flowchart. Qualification of patients to the study is presented in good details, also regarding the inclusion/exclusion criteria.
Presentation of data is clear and does not raise major comments.
The discussion is well performed with correctly chosen references.
Comments:
-As indicated in the final conclusions it would be worth to look at the oxLDL level, but also 8-OHdG / oxodG.
- Some typing errors should be removed before publication of the manuscript, e.g. page 2, line 51 and line 60.
Author Response
Together with the co-authors, we thank the reviewers for their careful reading and constructive comments. We present here the revised form of our manuscript entitled “Antioxidant efficacy of a spirulina liquid extract on oxidative stress status and metabolic disturbances in subjects with metabolic syndrome”.
We have taken all the reviewers' comments into account and believe that the revised version is significantly improved and will be of interest to readers.
Reviewer3
-As indicated in the final conclusions it would be worth to look at the oxLDL level, but also 8-OHdG / oxodG.
We agree with this interesting comment. Indeed, 8-OHdG / oxodG is a good a biomarker of oxidative DNA damage. As we are planning another trial with more subjects, a higher dose of Spirulysat® and a longer supplementation period, we will introduce the measurement of this parameter.
- Some typing errors should be removed before publication of the manuscript, e.g., page 2, line 51 and line 60. We corrected this error, Done

Reviewer 4 Report
Very interesting and relevant topic, but with some flaws that need a reformulation.
An update should be made to the all Spirulina text for Arthrospora, after correction in lines 58 and 59.
Line 58 and 59 – Adapt the text, second the nomenclature - the first time it appears in the text e.g. Arthrospora platensis (formely Spirulina platensis) from phylum Cyanobacteria
Several inaccuracies throughout the text must be repaired as some of the examples presented
Line 55 pirulina? Or Spirulina? – confirm
Line 61 [17].
The bioactive activity of Spirulysat® should be presented more concretely. Which metabolic pathways are affected. What are bioactive molecules? Has the molecular components of Spirulysat® been studied?
Methodology
Why is Spirulysat® taken at breakfast?
What recommendations were made before blood and urine collections. Study of lipid profile was done an 8h fasting? Was 24-hour urine collected? Blood was collected in a tube with or without anticoagulant.
What methodologies are applied to assess biological parameters? Nothing is mentioned, only the laboratory.
The content is interesting but needs to be revised and corrected
Author Response
Together with the co-authors, we thank the reviewers for their careful reading and constructive comments. We present here the revised form of our manuscript entitled “Antioxidant efficacy of a spirulina liquid extract on oxidative stress status and metabolic disturbances in subjects with metabolic syndrome”.
We have taken all the reviewers' comments into account and believe that the revised version is significantly improved and will be of interest to readers.
Reviewer 4
Very interesting and relevant topic, but with some flaws that need a reformulation.
An update should be made to the all Spirulina text for Arthrospora, after correction in lines 58 and 59.
Line 58 and 59 – Adapt the text, second the nomenclature - the first time it appears in the text e.g. Arthrospora platensis (formely Spirulina platensis) from phylum Cyanobacteria Line 44
Several inaccuracies throughout the text must be repaired as some of the examples presented
Line 55 pirulina? Or Spirulina? – confirm Line 45
- We thank the reviewers for their comments and apologise for any errors left in the text. We have taken into account the updated nomenclature throughout the text : An update should be made to the all Spirulina text for Arthrospora, after correction in lines 58 and 59. Done
- Line 58 and 59 – Adapt the text, second the nomenclature - the first time it appears in the text e.g., Arthrospora platensis (formely Spirulina platensis) from phylum Cyanobacteria
Several inaccuracies throughout the text must be repaired as some of the examples presented Done
- Line 55 pirulina? Or Spirulina? – Done
- Line 61 [17]. Done
- The bioactive activity of Spirulysat® should be presented more concretely. Which metabolic pathways are affected. What are bioactive molecules? Has the molecular components of Spirulysat® been studied?
Spirulysat® is a liquid extract of fresh Spirulina (Arthrospira platensis) titrated in C-PC. This extract contains, in addition to phycocyanin C-PC, other water-soluble molecules, proteins, amino acids, enzymes, sugars, water-soluble vitamins and mineral salts. Line 210 to line 211
The observed results in the present trial are probably related to a synergic effect of these molecules and essentially to a predominant antioxidant component, the phycocyanin. It is well established that oxidative stress and inflammation are involved in the development of cardiovascular disease [1,2]. Phycocyanin, the main component of Spirulysat® exhibits an antioxidative activity because it is able to scavenge various radicals and to inhibit lipid peroxidation [3,4]. In addition, phycocyanin has anti-inflammatory properties [5,6]. In our previous studies in Hamster and mice we observed a strong association between Spirulysat® supplementation and antioxidant parameters, bile acid modification, impact on food intake, modulation of gene expression. Since in the present trial we measured a decrease in urinary isoprostanes, we can suggest that this antioxidative effect observed with Spirulysat® leads to ameliorating lipids profile (decrease in plasma triglycerides and increase in HDL cholesterol) we have measured.These precisions are introduced in the disscussion section, Line 148 to line 152 and line 157 to line 161
Methodology
-Why is Spirulysat® taken at breakfast?
It is possible that taking Spirulysat® with or away from a meal will not lead to the same results. However, in the present study, we defined the time of intake as breakfast without any supporting scientific data, and only so that all participants take it at the same time to avoid any variability related to the time of intake.
This sentence is introduced in the manuscript « The choice of taking the Spirulysat at breakfast was done without any supporting scientific data, and only so that all participants take it at the same time to avoid any variability related to the time of intake ». Line 260 to Line 261
-What recommendations were made before blood and urine collections. Study of lipid profile was done an 8h fasting? Was 24-hour urine collected? Blood was collected in a tube with or without anticoagulant, what methodologies are applied to assess biological parameters?
We thank the reviewer for these comments and apologize for the lack of methods for assessing the biological parameters that we introduce in the revised manuscript.
All biological parameters were analyzed in a central laboratory (Biofortis Mérieux Nutrisciences). Blood and urine samples were taken after a 12-hour fasting period. The first one at visit 0 (Screening), the second one at visit 1 (Inclusion from 7 to 15 days after V0), the third one V2 (Follow up visit 15±3 after V1), the fourth one V3 end of study visit (15±3 days after V2). The blood is collected in different tubes depending on the dosage to be performed. For glucose determination the Fluor tubes were used and analysed by enzymatic Hexokinase/Roche diagnostic. Dry serum tubes were used for lipids determination. Concentration of total cholesterol and triglycerides were determined using enzymatic kit (Cobas integra 400+) anand that HLD-c by Photometry (Roche diagnostic) for. The liver markers Alanine aminotransferase (ALAT), Aspartate aminotransferase (ASAT) and gammaglutamyl-transferases (GGT) were analysed by UV test (Cobas integra 400+/Roche diagnostic), and GGT by Enzymatic (Roche diagnostic). LDLc was calculated according to friedewald formula, and LDLox by ELISA technique. For the measure of urinary isoprostane, all urine samples were collected 24 hours prior to each visit (24 hours prior to visit V1, 24 hours prior to visit V2 and 24 hours prior to visit V3) and the assay was performed only on the 24-hour urine. The urine was homogenised before aliquoting, then each aliquot was treated with Butylhydroxytoluene (BHT) (1mL of urine with 10µL of 0.025M BHT solution) before freezing for analysis. Urinary isoprostane (isoprostane F2 alpha) was analysed by ELISA (Oxford Biomedical Research). The raw results from the laboratory were expressed in ng/mL and then converted to the concentration in the whole 24h collected urine (µg/24h).
This paragraph is introduced in section method, Line 319 to line 334
What recommendations were made before blood and urine collections?
Pre-study recommendations:
All participants were given the following instructions: not to change lifestyle habits during the study (physical activity, smoking and alcohol consumption), not to do strenuous exercise during the 2 days preceding each visit, their eating habits during the study (no special diet, no taking of food supplements, no heavy meals or alcohol abuse), not to donate blood chronic drug treatments for those authorized (active and dosage) and not to take any new drugs during the study (having an impact on the study parameters), except in extreme cases.
This paragraph is introduced in the revised manuscript. Line 235 to line 240
References
- Jürgens, G.; Hoff, H.F.; Chisolm, G.M.; Esterbauer, H. Modification of Human Serum Low Density Lipoprotein by Oxidation--Characterization and Pathophysiological Implications. Chem. Phys. Lipids 1987, 45, 315–336, doi:10.1016/0009-3084(87)90070-3.
- Libby, P. Inflammation in Atherosclerosis. Arterioscler. Thromb. Vasc. Biol. 2012, 32, 2045–2051, doi:10.1161/ATVBAHA.108.179705.
- Romay, C.; Ledón, N.; González, R. Further Studies on Anti-Inflammatory Activity of Phycocyanin in Some Animal Models of Inflammation. Inflamm. Res. Off. J. Eur. Histamine Res. Soc. Al 1998, 47, 334–338, doi:10.1007/s000110050338.
- Thangam, R.; Suresh, V.; Asenath Princy, W.; Rajkumar, M.; Senthilkumar, N.; Gunasekaran, P.; Rengasamy, R.; Anbazhagan, C.; Kaveri, K.; Kannan, S. C-Phycocyanin from Oscillatoria Tenuis Exhibited an Antioxidant and in Vitro Antiproliferative Activity through Induction of Apoptosis and G0/G1 Cell Cycle Arrest. Food Chem. 2013, 140, 262–272, doi:10.1016/j.foodchem.2013.02.060.
- Romay, C.; Armesto, J.; Remirez, D.; González, R.; Ledon, N.; García, I. Antioxidant and Anti-Inflammatory Properties of C-Phycocyanin from Blue-Green Algae. Inflamm. Res. Off. J. Eur. Histamine Res. Soc. Al 1998, 47, 36–41, doi:10.1007/s000110050256.
- Romay, C.; González, R.; Ledón, N.; Remirez, D.; Rimbau, V. C-Phycocyanin: A Biliprotein with Antioxidant, Anti-Inflammatory and Neuroprotective Effects. Curr. Protein Pept. Sci. 2003, 4, 207–216, doi:10.2174/1389203033487216.

Round 2
Reviewer 1 Report
The authors have addressed all my concerns in the revised manuscript. I support the publication of the revised manuscript.
Reviewer 2 Report
This revised manuscript can be accepted. No further comments.
Reviewer 3 Report
The comments have been discussed.
Reviewer 4 Report
See “algaeBase.org”
valid name Arthrospira
valid name is Arthrospira and not Spirulina, review all text
Spirulina or Arthrospira or Arthrospira platensis - always in italics and capital letters (taxonomy rules)
First time it appears in the text, Arthrospira (formely Spirulina), then apply the valid name
Line 17 - such as Arthrospira (formely Spirulina), helps …
Line 18 – the Arthrospira liquid ….
Line 43 – Arthrospira (formerly Spirulina)
Line 44 - Arthrospira platensis of the phylum Cyanobacteria grows in aquatic ecosystems
Line 47, 48 - Arthrospira platensis, Arthrospira maxima and Arthrospira fusiformis [14]. Arthrospira platensis which ….
Line 130 – compared to a placebo on the level of oxidized LDL, glucose, metabolism lipid metabolism and 24 hour urinary isoprostanes in subjects with metabolic syndrome.
Line 142 – in vitro
Line 146 - fresh Arthrospira platensis
Line 160 - Arthrospira [22,34,35] and it …
Line 161 - … in the liquid extract isolated from Arthrospira platensis
Line 206 – subtitle Arthrospira…
Line 209 and 210 – g/L (standardize the units)
Line 282 – Fig 1 Blood collection
Table 4 – of no interest, repeats the results presented in tab 2 (V1)
Line 317 and 318 - Blood samples were taken after a 12-hour fasting period. Urine was collected for 24 h and after mixing a sample was taken for analysis.
Line 327 and 328 - samples were collected during 24 hours prior to each visit (V1, V2, V3) and the assay was performed only on the 24-hour urine to take into account the difference in the dilution 328 of participant’s urine.
Good work!